# Association between COVID-19 vaccination and sudden death in apparently healthy younger individuals: A population-based case-control study

Husam Abdel-Qadir[1,2,3☯*], Hardil Anup Bhatt[4,5☯], Sarah Swayze[3],
Michael Paterson[3,6,7], Dennis T. Ko[3,7,8,9], David N. Juurlink[3,10], Jeffrey C. Kwong[3,11,12,13,14,15]

1 Women's College Hospital, Toronto, Ontario, Canada, 2 Peter Munk Cardiac Centre, University Health Network, Toronto, Ontario, Canada, 3 ICES, Toronto, Ontario, Canada, 4 Temerty Faculty of Medicine, University of Toronto, Toronto, Ontario, Canada, 5 Department of Medicine, University of British Columbia, Vancouver, British Columbia, Canada, 6 Department of Family Medicine, McMaster University, Hamilton, Ontario, Canada, 7 Institute of Health Policy, Management and Evaluation, University of Toronto, Toronto, Ontario, Canada, 8 Department of Medicine, University of Toronto, Toronto, Ontario, Canada, 9 Schulich Heart Centre, Sunnybrook Research Institute, Sunnybrook Health Sciences Centre, Toronto, Ontario, Canada, 10 Department of Medicine, Sunnybrook Health Sciences Centre, University of Toronto, Toronto, Ontario, Canada, 11 Dalla Lana School of Public Health, University of Toronto, Toronto, Ontario, Canada, 12 Public Health Ontario, Toronto, Ontario, Canada, 13 University Health Network, Toronto, Ontario, Canada, 14 Department of Family and Community Medicine, University of Toronto, Toronto, Ontario, Canada, 15 Centre for Vaccine Preventable Diseases, University of Toronto, Toronto, Ontario, Canada

☯ These authors contributed equally to this work.
* h.abdel.qadir@utoronto.ca

## Abstract

### Background

COVID-19 vaccines can cause rare but serious adverse events such as myocarditis and immune thrombotic thrombocytopenia. Despite a lack of strong evidence, concerns have been expressed that COVID-19 vaccination might lead to sudden death in younger healthy adults. We studied the association between COVID-19 vaccination and sudden death in apparently healthy people aged 12–50 years.

### Methods and findings

We conducted a population-based case-control study using linked administrative datasets of residents of Ontario, Canada who were alive as of April 1, 2021. We excluded individuals aged >50 years and those with documented cardiovascular disease, mental illness, or diseases that predispose to adverse outcomes from COVID-19. We defined cases as those with out-of-hospital death, or death within 24 hours of presentation to hospital with a final diagnosis of cardiac arrest between April 1, 2021 and June 30, 2023. We matched each case with five

**Data availability statement:** The dataset from this study is held securely in coded form at ICES. While legal data sharing agreements between ICES and data providers (e.g., healthcare organizations and government) prohibit ICES from making the dataset publicly available, access may be granted to those who meet pre-specified criteria for confidential access, available at http://www.ices.on.ca/DAS (email: das@ices.on.ca). The full dataset creation plan and underlying analytic code are available from ICES upon request, understanding that the computer programs may rely upon coding templates or macros that are unique to ICES and are therefore either inaccessible or may require modification. To view details on the types of data and datasets available, please visit the ICES Data Dictionary (at https://datadictionary.ices.on.ca/Applications/DataDictionary/Default.aspx).

**Funding:** This work was supported by funding from the Canadian Immunization Research Network (CIRN, [https://landing.cirnetwork.ca]) through a grant from the Public Health Agency of Canada and the Canadian Institutes of Health Research (CNF 151944), and funding from the Public Health Agency of Canada, through the Vaccine Surveillance Working Party and the COVID-19 Immunity Task Force (both grants to JCK). This study was supported by Public Health Ontario and by ICES, which is funded by an annual grant from the Ontario Ministry of Health (MOH) and Ministry of Long-Term Care (MLTC). This work was also supported by the Ontario Health Data Platform (OHDP), a Province of Ontario initiative to support Ontario's ongoing response to COVID-19 and its related impacts. HA-Q is supported by a Tier 2 Canada Research Chair in Cardiovascular Disease Epidemiology and Outcomes (from the Government of Canada), a Hold'em for Life Professorship in Cancer Research (from the University of Toronto) and was previously supported by a Chair in Women's Heart and Brain Health (from the Heart and Stroke Foundation of Canada). JCK is supported by a Clinician-Scientist Award from the University of Toronto Department of Family and Community Medicine. The funders had no role in study design, data collection and analysis, decision to publish, or preparation of the manuscript.

controls on age, sex, region of residence, and neighborhood income quintile. We used conditional logistic regression to assess the association between sudden death and previous COVID-19 vaccination after adjusting for multiple potential confounders (positive severe acute respiratory syndrome coronavirus 2 [SARS-CoV-2] tests, number of SARS-CoV-2 polymerase chain reaction (PCR) tests, influenza vaccination, common comorbidities). Sensitivity analyses were conducted with different definitions of the exposure and subsets of cases (with their matched controls). Another sensitivity analysis utilized a modified self-controlled case series (SCCS) of vaccinated individuals meeting the case definition during the study period with up to three doses of any COVID-19 vaccine.

Of 6,365,451 eligible individuals, we identified 4,963 (0.08%) cases meeting our definition of sudden death (median age 36 years, 74.4% male). In the primary analysis, COVID-19 vaccination was associated with a lower risk of sudden death (adjusted odds ratio [aOR] = 0.57; 95% confidence interval (CI) [0.53,0.61]; $p < 0.001$). The findings were consistent for COVID-19 vaccination within six weeks before death (aOR = 0.63; 95%CI [0.55,0.72]; $p < 0.001$) and in sensitivity analyses limited to people aged <40 years (aOR = 0.53; 95%CI [0.48,0.58]; $p < 0.001$), those who died in hospital or in the emergency department (aOR = 0.71; 95%CI [0.55,0.91]; $p = 0.006$), and after exclusion of opioid-related deaths (aOR = 0.57; 95%CI [0.51,0.64]; $p < 0.001$). The SCCS sensitivity analysis showed no significant difference in the rate of sudden death in the 6 weeks following first (relative incidence (RI) 0.87; 95%CI [0.54,1.40]; $p = 0.57$), second (RI 0.94; 95%CI [0.57,1.57]; $p = 0.82$), or third (RI 0.87; 95%CI [0.37,2.05]; $p = 0.10$) dose of the COVID-19 vaccine. Study limitations include the inability to confirm the cause of out-of-hospital deaths and residual confounding due to differences in health-seeking behaviors for the case-control analysis.

## Conclusions

These findings do not support the hypothesis that COVID-19 vaccines increase the risk of sudden cardiac death in young healthy adults.

## Author summary
### Why was this study done?

- COVID-19 vaccines were received by a large segment of the population as part of the public health response to the pandemic

- There are emerging concerns that COVID-19 vaccines are responsible for sudden death in younger healthy individuals despite a lack of evidence to support this claim

**Competing interests:** The authors have declared that no competing interests exist.

**Abbreviations:** aOR, adjusted odds ratio; CI, confidence interval; CIHI, Canadian Institute for Health Information; COPD, chronic obstructive pulmonary disease; CVD, cardiovascular disease; DAD, Discharge Abstract Database; ED, emergency department; HIV, human immunodeficiency virus; mRNA, messenger ribonucleic acid; NACRS, National Ambulatory Care Reporting System; OHIP, Ontario Health Insurance Plan; PCR, polymerase chain reaction; PHIPA, Personal Health Information Protection Act; RECORD, Reporting of Studies Conducted using Observational Routinely-Collected Data; RI, relative incidence; SARS-CoV-2, severe acute respiratory syndrome coronavirus 2; SCCS, self-controlled case series; VITT, vaccine-induced thrombotic thrombocytopenia.

## What did the researchers do and find?

- A case-control study was conducted involving Ontario residents aged 12–50 years without documented comorbidities predisposing to premature death between April 1, 2021 and June 30, 2023 to examine the association between COVID-19 vaccination and sudden death

- The primary outcome was sudden death; the exposure of interest was any COVID-19 vaccination

- Among 6,365,451 eligible individuals, 4,806 cases who experienced sudden death were matched to 24,030 controls who were alive on the date of sudden death for each corresponding case

- Receipt of COVID-19 vaccination was not associated with increased odds of sudden death

## What do these findings mean?

- These findings do not support the hypothesis that COVID-19 vaccines increase the risk of sudden cardiac death in younger healthy adults

- A limitation of this study was the inability to confirm the cause of out-of-hospital deaths

## Introduction

Messenger ribonucleic acid (mRNA) vaccines were an essential element of the strategy for curtailing the COVID-19 pandemic [1,2]. Emergency approval of the vaccines was provided in an accelerated timeframe after large-scale clinical trials demonstrated robust efficacy and relatively benign adverse event profiles. With widespread use, however, it became clear that mRNA vaccines were associated with cases of myocarditis, most prominently in males aged <40 years [3]. The non-replicating viral vector vaccines were associated with vaccine-induced immune thrombotic thrombocytopenia (VITT), which led to rare cases of fatal thromboembolism [4]. Despite these concerns, the anticipated benefits of these vaccines were deemed to outweigh their risks at the time, and more than 75% of the adult population in high-income countries have been vaccinated [5].

Recently, concerns have been raised that healthy younger people were dying suddenly because of COVID-19 vaccination, however, this notion is not supported by any reliable scientific evidence [6–9]. Because most people living in the Western hemisphere have been vaccinated, most sudden deaths are expected to occur in previously vaccinated people. It is plausible that subclinical myocarditis after vaccination could predispose to arrhythmias if residual scar tissue served as an arrhythmogenic focus. In a prospective evaluation of 54 participants, we previously demonstrated that 1 in 8

people with confirmed acute symptomatic myocarditis had evidence of focal myocardial inflammation by fluorodeoxyglucose positron emission tomography/magnetic resonance imaging at two months follow-up [10]. A longer-term follow-up study in 13 patients with symptomatic myocarditis using cardiac MRI demonstrated that myocardial edema had resolved in all participants, although small areas of myocardial scar persisted in 2 (13%) patients [11].

There are limited data on the potential association of COVID-19 vaccination with sudden death in healthy people. Concerns about sudden death due to COVID-19 vaccination can deter future vaccination [12]. Accordingly, we conducted a population-based case-control study to explore the hypothesis that COVID-19 vaccination increases risk of sudden death in apparently healthy people aged 12–50 years.

## Methods

### Study design, setting, and population

We used a case-control design and linked databases in the Canadian province of Ontario, which provides universal health coverage to all residents through the Ontario Health Insurance Plan (OHIP). The datasets were linked using unique encoded identifiers and analyzed at ICES (previously known as the Institute for Clinical Evaluative Sciences [www.ices.on.ca]) [13,14]. ICES is an independent, non-profit research institute whose legal status under Ontario's health information privacy law allows it to collect and analyze healthcare and demographic data, without consent, for health system evaluation and improvement.

We used Ontario's Registered Persons Database to identify all residents who were alive on April 1, 2021. This date was chosen to allow for accrual during a period with fewer restrictions on vaccine access for healthier people. We excluded residents without OHIP eligibility on April 1, 2021, those with no recorded contact with the healthcare system in the prior 10 years (to decrease the potential for undocumented disease), those missing key data (date of birth, sex, postal code), and those younger than 12 years (given lower vaccine coverage and weaker vaccination mandates).

We then applied exclusion criteria to limit the study population to people who did not have diseases that would predispose them to sudden cardiovascular death or adverse outcomes from COVID-19, and those with documented mental illness (excluding mood/anxiety disorders) [15,16]. Exclusions included age 50 years or older, long-term care residence, schizophrenia [17], healthcare encounters for alcohol or illicit drug use in the 5 years preceding April 1, 2021 [18], cardiovascular disease (CVD; including coronary artery disease, heart failure, and atrial fibrillation) [19], diabetes [20], cancer [21], dementia [22], chronic obstructive pulmonary disease (COPD) [23], chronic liver or kidney disease [24,25], inflammatory bowel disease [26], autoimmune rheumatologic disease [27,28], human immunodeficiency virus (HIV) infection [29], frailty (as per the Johns Hopkins ACG System Version 10 frailty indicator) [30], other forms of immunocompromised status/autoimmune diseases [31], or receipt of chronic home care in the 5 years before April 1, 2021. We did not exclude people with hypertension, asthma, or mood/anxiety disorders given the high prevalence of these diagnoses in otherwise healthy people. Application of these exclusions left us with a study population aged 12–50 years without documented illnesses that are expected to be strongly associated with either the exposure (COVID-19 vaccination) or the outcome of interest (death).

### Definition of cases and controls

Cases were required to fulfill one of three criteria between April 1, 2021 and June 30, 2023:

1. Death outside of hospital; OR

2. Death in an emergency department (ED), with a most responsible discharge diagnosis of cardiac arrest, sudden death, or significant ventricular arrhythmia, and where none of the other diagnostic codes indicated trauma, mental illness, or substance use (see S1 Table); OR

3. In-hospital death within 24 hours of admission, with a most responsible discharge diagnosis of cardiac arrest, sudden death, or ventricular arrhythmia, and where no other diagnostic code indicated trauma, mental illness, or substance use (see S1 Table)

PLOS Medicine

We ascertained hospitalization status and discharge diagnoses from the Canadian Institute for Health Information (CIHI) Discharge Abstract Database (DAD), and ED records from the CIHI National Ambulatory Care Reporting System (NACRS).

We matched each case with up to 5 controls from the pool of eligible individuals who were alive on the index date, matching on age, sex, geographic area of residence (based upon the forward sortation area), and neighborhood income quintile. The index date for controls was the date of death of their matched case.

## Definition of the exposure

The exposure of interest was receipt of any COVID-19 vaccination, which was ascertained from the COVaxON database, a centralized registry that identifies all COVID-19 vaccination events in Ontario [32].

## Statistical analysis

The primary and secondary sensitivity case-control analyses were prospectively planned on January 5, 2023. The post-hoc analysis with modified self-controlled case series (SCCS) was performed after the peer-review process to address potential confounding associated with health-seeking behaviors of vaccine recipients.

Baseline characteristics of case and controls were summarized using medians (with 25th–75th percentiles) for continuous variables and counts with percentages for categorical variables. Given the large sample size, we used standardized differences rather than p-values to compare baseline differences between groups, with values greater than 0.1 indicating meaningful differences [33].

We used conditional logistic regression to determine the association between COVID-19 vaccination and the odds of being a case or a control while accounting for the matched nature of the sample. The model included terms for a positive SARS-CoV-2 test in the preceding 90 days (since SARS-CoV-2 infection can predispose to myocarditis [34]), a positive SARS-CoV-2 test >90 days prior, influenza vaccination in the preceding 365 days, and the number of SARS-CoV-2 PCR tests in the preceding year (to adjust for health-seeking behaviors), as well as a history of asthma, hypertension, and mood or anxiety disorders.

We conducted several sensitivity analyses to test the robustness of our conclusions. We first limited the exposure to mRNA vaccines, given their association with myocarditis, and then the AstraZeneca vaccine, given its association with thromboembolic events. We next examined COVID-19 vaccination in the six weeks preceding the index date (the period when myocarditis is most likely). We then limited cases to deaths in hospital or the ED where the cause of death was documented to be due to sudden cardiac death, with exclusion of diagnostic codes indicating trauma, mental illness, or substance use. Furthermore, we restricted the analysis to people aged 40 years or younger, a population with a lower likelihood of undiagnosed cardiac disease and a higher risk of vaccine-associated myocarditis. In addition, we repeated the analysis with a secondary case definition that excluded opioid-related deaths using the Drug and Drug/Alcohol Related Death (DDARD) database. This analysis was limited to deaths before June 30, 2022 due to data availability.

In an additional post-hoc sensitivity analysis, we utilized a modified SCCS method [35,36]. This was restricted to vaccinated residents meeting the case definition during the study period who had up to three doses of any COVID-19 vaccine. For this analysis, the observation period remained the same (April 1, 2021 to June 30, 2023). We used a 6-week risk interval beginning on the vaccination date (day 0) for each dose received, with person-time for each interval included in the model as an offset. The control period was the remainder of the observation period after exclusion of the 6-week risk intervals if applicable. Vaccine exposures were modeled as first dose, second dose, or third dose. Relative incidence (RI) was estimated using conditional Poisson regression with a pseudo-likelihood approach and sandwich variance estimators to calculate Wald confidence intervals. Models were adjusted for age as a time-varying covariate and for quarterly rate of sudden death among unvaccinated individuals derived from our case-control analysis.

All analyses were conducted using SAS Version 9.4 (SAS Institute). Statistical significance for comparisons of outcomes was defined as a two-tailed *p*-value <0.05. This study is reported as per the Reporting of Studies Conducted using Observational Routinely-Collected Data (RECORD) guideline (S1 Checklist).

## Ethics approval

The use of most data in this project is authorized under section 45 of Ontario's Personal Health Information Protection Act (PHIPA) and does not require review by a Research Ethics Board [37]. Ethics approval was required for the analyses excluding opioid-related deaths, which was part of our secondary case definition; this was obtained from the Women's College Hospital Research Ethics Board (REB # 2024-0002-E).

## Results

We identified 14,664,193 Ontario residents who were alive as of April 1st, 2021. After applying exclusion criteria (Fig 1), we were left with 6,365,451 eligible individuals. During the accrual period, 4,963 (0.08%) individuals died and met criteria to be cases. Of these, 4,448 (89.6%) died in the prehospital setting and 515 (10.4%) died within 24 hours of presenting to a hospital or ED with a discharge diagnosis of cardiac arrest. From this pool of 4,963 cases, 4,806 (96.8%) were included in the primary case-control analysis after each was matched to 5 controls (for a total of 24,030 controls).

Prior to matching, cases were older and were more likely to be male compared to controls. They were also more likely to reside in northern Ontario, and less likely to reside in Toronto or the neighboring regions of Peel and York (S2 Table). Cases were more likely to live in neighborhoods with lower income, fewer people per household, lower proportions of visible minorities, and higher proportions of people employed in sales, trades, manufacturing, or agriculture (industries more likely to stay open during pandemic-related lockdowns). There was a higher prevalence of hypertension and mood/anxiety disorders, but a lower prevalence of documented influenza vaccination among cases. There were no relevant differences in the percentage receiving COVID-19 vaccination. A total of 4,806 cases were matched with 5 controls on age, sex, and forward sortation area (Table 1). Most differences between cases and controls were nullified after matching, but there remained a higher prevalence of hypertension and mood/anxiety disorders and lower documented COVID-19 and influenza vaccinations among cases.

A total of 3,237 (67.4%) of cases had prior COVID-19 vaccination, compared with 18,520 (77.1%) of controls. Two or more COVID-19 vaccines had been received before the index date in 2,912 (60.6%) cases and 17,097 (71.1%) controls. When limiting the exposure to mRNA vaccines, at least one dose was documented in 3,212 (66.8%) cases compared to 18,374 (76.5%) controls. These observations were mostly driven by the Pfizer/BioNTech Comirnaty vaccine, which was received by 2,482 (51.6%) cases and 14,801 (61.6%) controls. The standardized differences for all these comparisons were ≥0.1. For vaccines that were administered in lower numbers, such as Moderna's Spikevax and AstraZeneca's Vaxzevria vaccines, the prevalence of vaccination was low in both groups. There were fewer than 6 cases who died with prior exposure to Janssen's Jcoyden vaccine and none with exposure to Novavax's Nuvaxovid vaccine. When we specifically focused on COVID-19 vaccination in the preceding six weeks, vaccination was documented for 317 (6.6%) of cases compared with 2,246 (9.3%) of controls (standardized difference = 0.1).

In the primary analysis, COVID-19 vaccination was associated with lower odds of sudden death (adjusted odds ratio [aOR] = 0.57; 95%CI [0.53,0.61]; *p*<0.001; Fig 2). Documented influenza vaccination was also associated with lower odds of death (aOR = 0.72; 95%CI [0.66,0.80]; *p*<0.001). A documented positive SARS-CoV-2 PCR test within 90 days of the index date was associated with higher odds of death (aOR = 2.36; 95%CI [1.84,3.02]; *p*<0.001), while a positive SARS-CoV-2 PCR test greater than 90 days before the index date was associated with lower odds of death (aOR = 0.83; 95%CI [0.72,0.95]; *p* = 0.006). The odds of death were also increased by the presence of asthma (aOR = 1.26; 95%CI [1.16,1.36]; *p*<0.001), hypertension (aOR = 1.70; 95%CI [1.50,1.92]; *p*<0.001), and a mood or anxiety disorder (aOR = 3.46; 95%CI [2.94,4.07]; *p*<0.001).

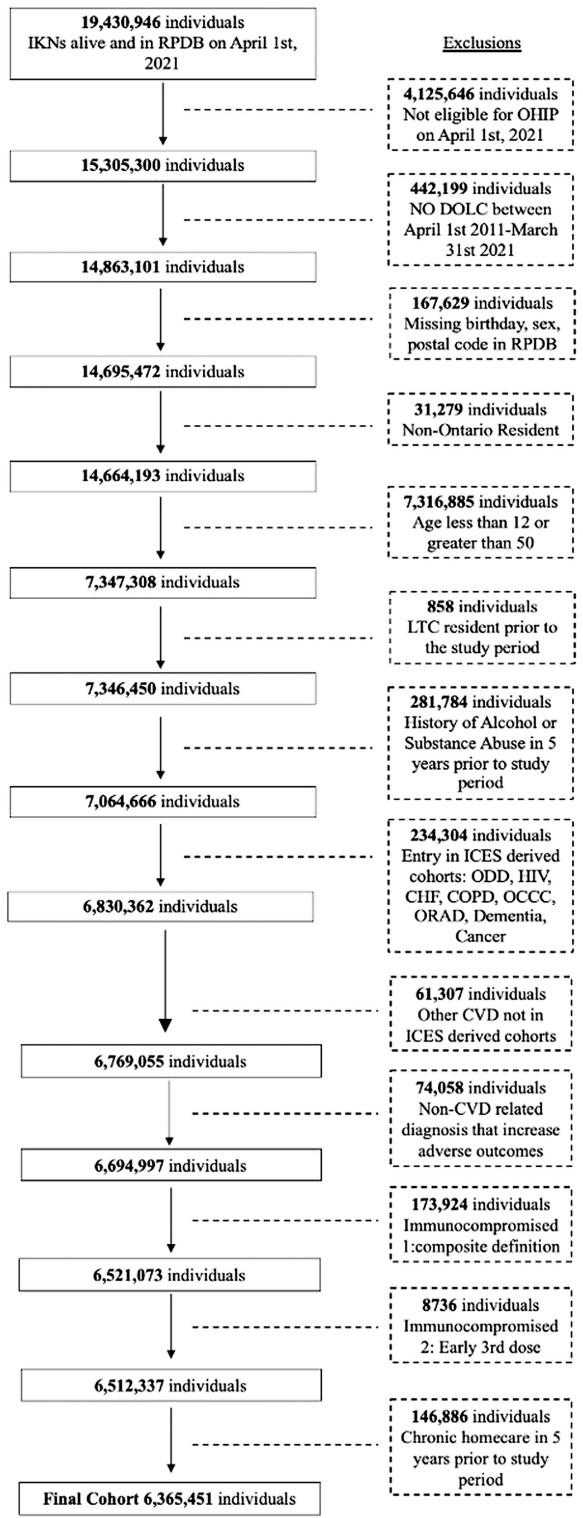

**Fig 1. Selection criteria of the study cohort.** CHF, Congestive Heart Failure; COPD, Chronic Obstructive Pulmonary Disease; CVD, cardiovascular disease; Dementia, Ontario Dementia Database; HIV, Ontario HIV Database; IKN, ICES key number; LTC, long term care; OCCC, Ontario Crohn's and Colitis Cohort dataset; ODD, Ontario Diabetes Dataset; OHIP, Ontario Health Insurance Plan Claims Database; ORAD, Ontario Rheumatoid Arthritis Database; RPDB, Registered Persons Database.

**Table 1. Baseline characteristics of residents of Ontario, Canada who were included in the case-control analysis.**

| Variable | Cases | Controls | Std. Diff* |
|---|---|---|---|
| | *N*=4,806 | *N*=24,030 | |
| **Age, mean ± SD, years** | 35.21 ± 10.40 | 35.21 ± 10.39 | 0 |
| **Age, median (IQR), years** | 36 (27-45) | 36 (27-45) | 0 |
| **Aged 12–18 years, *n*(%)** | 346 (7.2%) | 1,730 (7.2%) | 0 |
| **Aged 19–30 years, *n*(%)** | 1,309 (27.2%) | 6,545 (27.2%) | 0 |
| **Aged 31–40 years, *n*(%)** | 1,348 (28.0%) | 6,740 (28.0%) | 0 |
| **Aged 41–50 years, n(%)** | 1,803 (37.5%) | 9,015 (37.5%) | |
| **Male sex, *n*(%)** | 3,564 (74.2%) | 17,820 (74.2%) | 0 |
| **Public health unit region** | | | |
| Central East, *n*(%) | 384 (8.0%) | 1,863 (7.8%) | 0.01 |
| Central West, *n*(%) | 935 (19.5%) | 4,672 (19.4%) | <0.01 |
| Durham, *n*(%) | 237 (4.9%) | 1,176 (4.9%) | <0.01 |
| Eastern, *n*(%) | 340 (7.1%) | 1,779 (7.4%) | 0.01 |
| Northern, *n*(%) | 467 (9.7%) | 2,334 (9.7%) | <0.01 |
| Ottawa, *n*(%) | 301 (6.3%) | 1,477 (6.1%) | <0.01 |
| Peel, *n*(%) | 403 (8.4%) | 2,015 (8.4%) | <0.01 |
| Southwest, *n*(%) | 666 (13.9%) | 3,331 (13.9%) | <0.01 |
| Toronto, *n*(%) | 808 (16.8%) | 4,040 (16.8%) | <0.01 |
| York, *n*(%) | 255 (5.3%) | 1,293 (5.4%) | <0.01 |
| Missing data, *n*(%) | 10 (0.2%) | 50 (0.2%) | <0.01 |
| **Neighborhood income quintile** | | | |
| 1 (Lowest), *n*(%) | 1,291 (26.9%) | 6,455 (26.9%) | 0 |
| 2, *n*(%) | 991 (20.6%) | 4,955 (20.6%) | 0 |
| 3, *n*(%) | 946 (19.7%) | 4,730 (19.7%) | 0 |
| 4, *n*(%) | 837 (17.4%) | 4,185 (17.4%) | 0 |
| 5 (Highest), *n*(%) | 727 (15.1%) | 3,635 (15.1%) | 0 |
| Missing data, *n*(%) | 14 (0.3%) | 70 (0.3%) | 0 |
| **Neighborhood average number of persons per dwelling quintile** | | | |
| 1 (Lowest), *n*(%) | 1,011 (21.0%) | 4,801 (20.0%) | 0.03 |
| 2, *n*(%) | 968 (20.1%) | 4,636 (19.3%) | 0.02 |
| 3, *n*(%) | 605 (12.6%) | 3,213 (13.4%) | 0.02 |
| 4, *n*(%) | 1,018 (21.2%) | 5,233 (21.8%) | 0.01 |
| 5, *n*(%) | 860 (17.9%) | 4,384 (18.2%) | 0.01 |
| Missing, *n*(%) | 344 (7.2%) | 1,763 (7.3%) | 0.01 |
| **Neighborhood quintile by proportion of people who self-identify as visible minority quintile** | | | |
| 1 (Lowest), *n*(%) | 941 (19.6%) | 4,732 (19.7%) | <0.01 |
| 2, *n*(%) | 908 (18.9%) | 4,319 (18.0%) | 0.02 |
| 3, *n*(%) | 790 (16.4%) | 4,014 (16.7%) | 0.01 |
| 4, *n*(%) | 916 (19.1%) | 4,549 (18.9%) | <0.01 |
| 5 (Highest), *n*(%) | 906 (18.9%) | 4,653 (19.4%) | 0.01 |
| Missing, *n*(%) | 345 (7.2%) | 1,763 (7.3%) | 0.01 |
| **Neighborhood quintile by proportion employed in sales/trades/manufacturing/agriculture** | | | |
| 1 (Lowest), *n*(%) | 652 (13.6%) | 3,300 (13.7%) | 0 |
| 2, *n*(%) | 850 (17.7%) | 4,449 (18.5%) | 0.02 |
| 3, n(%) | 907 (18.9%) | 4,636 (19.3%) | 0.01 |

*(Continued)*

**Table 1.** (Continued)

| Variable | Cases | Controls | Std. Diff* |
|---|---|---|---|
| | *N* = 4,806 | *N* = 24,030 | |
| **4**, *n*(%) | 1,010 (21.0%) | 4,854 (20.2%) | 0.02 |
| **5 (Highest)**, *n*(%) | 1,042 (21.7%) | 5,028 (20.9%) | 0.02 |
| **Missing**, *n*(%) | 345 (7.2%) | 1,763 (7.3%) | 0.01 |
| Asthma, *n*(%) | 972 (20.2%) | 4,068 (16.9%) | 0.08 |
| Hypertension, *n*(%) | 411 (8.6%) | 1,335 (5.6%) | 0.12 |
| History of mood or anxiety disorder in the past 5 years, *n*(%) | 266 (5.5%) | 411 (1.7%) | 0.21 |
| Influenza vaccination in past year, *n*(%) | 587 (12.2%) | 4,024 (16.7%) | 0.13 |
| **Number of COVID-19 vaccine doses received as of index date** | | | |
| **0**, *n*(%) | 1,569 (32.6%) | 5,510 (22.9%) | 0.22 |
| **1**, *n*(%) | 325 (6.8%) | 1,423 (5.9%) | 0.03 |
| **≥2**, *n*(%) | 2,912 (60.6%) | 17,097 (71.1%) | 0.22 |
| Received any COVID-19 vaccine before index date, *n*(%) | 3,237 (67.4%) | 18,520 (77.1%) | 0.22 |
| Received COVID-19 vaccine within 6 weeks before index date, *n*(%) | 317 (6.6%) | 2,246 (9.3%) | 0.1 |
| Received ≥1 dose of any mRNA vaccine, *n*(%) | 3,212 (66.8%) | 18,374 (76.5%) | 0.21 |
| Received ≥1 dose of Pfizer/BioNTech Comirnaty vaccine, *n*(%) | 2,482 (51.6%) | 14,801 (61.6%) | 0.20 |
| Received ≥1 dose of Moderna Spikevax vaccine, *n*(%) | 1,458 (30.3%) | 7,999 (33.3%) | 0.06 |
| Received ≥1 dose of AstraZeneca Vaxzevria vaccine, *n*(%) | 154 (3.2%) | 1,055 (4.4%) | 0.06 |
| **Recent SARS-CoV-2 PCR test before case death date** | | | |
| **Never tested positive before**, *n*(%) | 4,406 (91.7%) | 22,119 (92.0%) | 0.01 |
| **Remote prior positive test (>90 days)**, *n*(%) | 293 (6.1%) | 1,694 (7.0%) | 0.04 |
| **Recent prior positive test (≤90 days)**, *n*(%) | 107 (2.2%) | 217 (0.9%) | 0.11 |
| **Number of SARS-CoV-2 PCR tests prior to case death date** | | | |
| **Mean ± SD** | 1.47 ± 3.61 | 1.18 ± 3.26 | 0.08 |
| **Median (IQR)** | 0 (0–2) | 0 (0–1) | 0.11 |

Note: The data are presented after dividing the cohort into two groups: cases and matched controls.

*std, standardized difference, IQR, interquartile range.

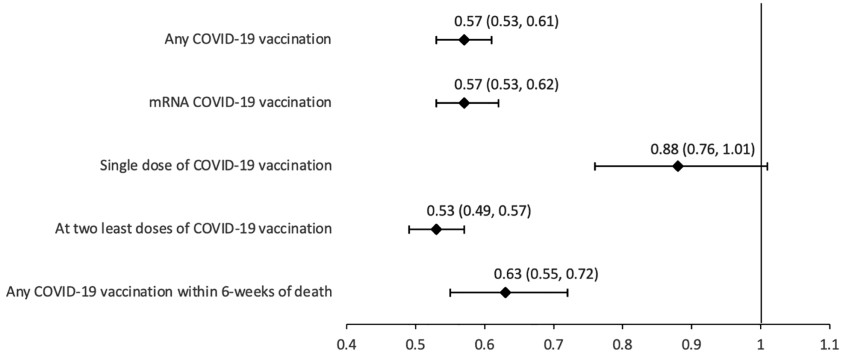

**Fig 2. The adjusted odds ratio (with 95% confidence interval) for death and COVID-19 vaccination among the matched cohort (*n* = 28,836) assessed using conditional logistic regression modeling.**

The inverse association of COVID-19 vaccination with death was consistent for most other definitions of vaccine exposure. Receipt of mRNA COVID-19 vaccination was associated with an aOR of 0.57 (95%CI [0.53,0.62]; $p < 0.001$) for death. Recent COVID-19 vaccination within six weeks prior to the index date was also associated with lower odds of death (aOR = 0.63; 95%CI [0.55,0.72]; $p < 0.001$). Individuals who received two doses of COVID-19 vaccination displayed a stronger negative association with death (aOR = 0.53; 95%CI [0.49,0.57]; $p < 0.001$), while receipt of only one dose was associated with lower risk of death that did not reach statistical significance (aOR = 0.88; 95%CI [0.76,1.01]; $p = 0.071$).

### People aged <40 years

The baseline characteristics of matched cases and controls in the subgroup aged <40 years are described in S3 Table. They demonstrated similar patterns to the overall cohort, wherein COVID-19 vaccination was less prevalent among cases (1,832 [64.3%]) compared to controls (10,778 [75.7%]). After adjusting for differences between cases and controls, there was an inverse association between any COVID-19 vaccination and sudden death (aOR = 0.53; 95%CI [0.48,0.58]; $p < 0.001$). The aORs for other definitions of COVID-19 vaccination are summarized in S1 Fig.

### Analyses excluding opioid-related deaths

The baseline characteristics of matched cases that excluded opioid-related deaths, along with matched controls, are described in S4 Table. Once again, COVID-19 vaccination was less prevalent among cases who died (1,232 [61.1%]) compared to living controls (7,159 [71.0%]). This pattern was observed in other definitions of the vaccination exposure, with standardized differences for all the comparisons being ≥0.1. After adjustment, we continued to observe an inverse association between any COVID-19 vaccination and sudden death (aOR = 0.57; 95%CI [0.51,0.64]; $p < 0.001$). The aORs for other definitions of COVID-19 vaccination are summarized in S2 Fig.

### Deaths in hospital or an emergency department

S5 Table lists the baseline characteristics of people who died within 24 hours of presenting to hospital or ED with a discharge diagnosis indicating cardiac arrest, with comparison to their matched controls. We observed a lower prevalence of COVID-19 vaccination among cases (365 [73.6%]) compared to controls (1,952 [78.7%]). Cases had lower proportions of people vaccinated with mRNA vaccines than controls (73.2% versus 78.1%) and a lower proportion vaccinated within 6 weeks prior to the index date (8.3% versus 11.4%). The standardized differences for all the comparisons described were ≥0.1. After adjustment, there continued to be an inverse association between COVID-19 vaccination and sudden death (aOR = 0.71; 95%CI [0.55,0.91]; $p = 0.006$). The aORs for other definitions of COVID-19 vaccination are summarized in S3 Fig.

### Modified SCCS analysis

The SCCS sensitivity analysis demonstrated that there was no significant difference in the rate of sudden death in the 6 weeks following first (RI 0.87; 95%CI [0.54,1.40]; $p = 0.57$), second (RI 0.94; 95%CI [0.57,1.57]; $p = 0.82$), or later doses (RI 0.87; 95%CI [0.37, 2.05]; $p = 0.10$) of a COVID-19 vaccine (S6 Table).

## Discussion

In this population-based study, vaccination against COVID-19 was not associated with an increased risk of sudden death in people younger than 50 years who had no documented evidence of cardiovascular disease. This finding persisted through sensitivity analyses limited to people aged <40 years, those who died in-hospital with a diagnosis of sudden cardiac arrest within 24 hours of presentation, after exclusion of admissions associated with trauma, mental illness, and substance use, after exclusion of opioid-related deaths, and another sensitivity analysis utilizing a modified SCCS. While

most Ontarians received the Pfizer/BioNTech mRNA vaccine, we did not observe a higher prevalence of exposure to any Health Canada-approved vaccine among people who died. Collectively, these observations refute the claim that COVID-19 vaccination increases the risk of sudden death.

Our data align with recent publications which report that COVID-19 vaccination is not associated with higher risk of sudden death. Previous studies from the United States and England utilized a modified SCCS design to demonstrate no significant increase in cardiac or all-cause mortality in the 4 weeks and 12 weeks after COVID-19 vaccination, respectively [38,39]. A case-control study from India suggested a decreased likelihood of sudden death in younger individuals who had received 2 or more doses of COVID-19 vaccines (odds ratio 0.58; 95%CI [0.37, 0.92]) compared to those who had received only 1 dose (odds ratio = 1.00; 95%CI [0.73,1.36]) [40]. Our primary case-control analyses similarly demonstrated a dose-dependent protective effect, while the SCCS sensitivity analysis showed no protective effect associated with any dose of COVID-19 vaccination. It is possible that the apparent protective effective of increasing doses of COVID-19 vaccination in the case-control analyses may reflect confounding due to greater health-seeking behaviors among vaccinated individuals.

While some studies have suggested dose-dependent safety concerns due to higher risks of myocarditis after the second dose [41,42], this does not align with our data and that from other large, epidemiologic analyses. Further supporting the absence of an increased risk of sudden death following COVID-19 vaccines, an analysis of mortality records of residents in Italy aged 1–40 years showed no increase in sudden cardiac death after the introduction of COVID-19 vaccines [43]. Indeed, rates of sudden death have decreased among athletes between the years 2002–2022, including the years when COVID-19 vaccines were introduced [44]. Furthermore, a study examining death certificates of younger individuals between the years 2021–2022 found no evidence directly attributing COVID-19 vaccination to cardiac death [45]. While mRNA vaccines have demonstrated an increased risk of myocarditis in younger men, mortality rates following the development of postvaccine myocarditis are lower than mortality rates following COVID-19-related myocarditis or conventional myocarditis [46]. Postmortem histopathologic analyses have raised potential causal relations between unexpected death and VITT following COVID-19 vaccination [47–49], but this may be less relevant for mRNA vaccines which comprise most COVID-19 vaccines in use today.

The strengths of our study include its large, population-based cohort of residents covered by a universal single-payer healthcare system. The combination of case-control and SCCS methods provides more confidence in our conclusions. A major limitation of this study is that we could not confirm the underlying cause of death out of hospital, so could not exclude deaths out of hospital that were due to motor vehicle collisions, violence, or suicides. However, we consistently found that COVID-19 vaccination was not associated with increased odds of deaths due to cardiac arrest diagnosed in hospital or the ED within 24 hours of presentation after exclusion of alternative causes of death. We could only utilize neighborhood-level data rather than individual data for income, proportion of visible minorities, and occupations. For SARS-CoV-2 testing, we accounted for PCR tests but could not capture results of rapid antigen tests. Moreover, the registries used to derive a cohort free of cardiac and other chronic diseases would not capture people with undiagnosed diseases. Lastly, differences in healthcare seeking behaviors may have led to residual confounding; this is expected to be less applicable for the SCCS analysis.

This population-based case-control study did not show a positive association between COVID-19 vaccination and death in apparently healthy individuals aged <50 years. We did not observe higher odds of death in people who received any of the Health Canada-approved COVID-19 vaccines. These data do not support the hypothesis that COVID-19 vaccines increase the risk of sudden cardiac death.

## Supporting information

**S1 Checklist. Reporting of Studies Conducted using Observational Routinely-Collected Data (RECORD) checklist.** (DOCX)

**S1 Table. ICD-10 diagnostic codes used for defining a case.**
(DOCX)

**S2 Table. Baseline characteristics of the residents of Ontario, Canada who met criteria for inclusion in the study.**
The data are presented after dividing the cohort into two groups—people who met the definition for being a case, and those who were eligible for being selected as controls. Please note that the cases are not matched to controls in this table.
(DOCX)

**S3 Table. Baseline characteristics of matched cases and controls aged <40 years.**
(DOCX)

**S4 Table. Baseline characteristics of matched cases and controls excluding opioid-related deaths.**
(S4_Table.DOCX)

**S5 Table. Baseline characteristics of matched cases and controls only including deaths within 24-hours of presenting to the hospital.**
(DOCX)

**S6 Table. The relative incidence (with 95% confidence interval) of sudden death in vaccinated individuals within the post COVID-19 vaccination risk period (6-weeks after the vaccination date for each dose received) compared to the control period.** The analysis adjusted for age and quarterly rate of sudden death among unvaccinated individuals.
(DOCX)

**S1 Fig. The adjusted odds ratio (with 95% confidence interval) for death and COVID-19 vaccination in individuals younger than 40 years of age among the matched cohort ($n$=17,094), assessed using conditional logistic regression modeling.**
(TIFF)

**S2 Fig. The adjusted odds ratio (with 95% confidence interval) for death and COVID-19 vaccination excluding opioid-related deaths among the matched cohort ($n$=12,096), assessed using conditional logistic regression modeling.**
(TIFF)

**S3 Fig. The adjusted odds ratio (with 95% confidence interval) for death and COVID-19 vaccination only involving deaths within 24-hours of arriving to the hospital among the matched cohort ($n$=2,976), assessed using conditional logistic regression modeling.**
(TIFF)

## Acknowledgments

We would like to acknowledge Public Health Ontario for access to vaccination data from COVaxON. This document used data adapted from the Statistics Canada Postal Code Conversion File, which is based on data licensed from Canada Post Corporation, and/or data adapted from the Ontario Ministry of Health Postal Code Conversion File, which contains data copied under license from Canada Post Corporation and Statistics Canada. Parts of this material are based on data and/or information compiled and provided by: the Ontario Ministry of Health, Ontario Health (OH), and CIHI. The analyses, conclusions, opinions, and statements expressed herein are solely those of the authors and do not reflect those of the funding or data sources. We thank IQVIA Solutions Canada for use of their Drug Information File. Importantly, we thank the Ontario residents, without whom this research would be impossible.

## Author contributions

**Conceptualization:** Husam Abdel-Qadir, Jeffrey C. Kwong.

**Data curation:** Husam Abdel-Qadir, Sarah Swayze, Michael Paterson, Dennis T. Ko, David N. Juurlink, Jeffrey C. Kwong.

**Formal analysis:** Husam Abdel-Qadir, Sarah Swayze, Michael Paterson, Dennis T. Ko, David N. Juurlink, Jeffrey C. Kwong.

**Funding acquisition:** Husam Abdel-Qadir, Jeffrey C. Kwong.

**Investigation:** Husam Abdel-Qadir, Hardil Anup Bhatt, Sarah Swayze, Michael Paterson, Dennis T. Ko, David N. Juurlink, Jeffrey C. Kwong.

**Methodology:** Husam Abdel-Qadir, Sarah Swayze, Michael Paterson, Dennis T. Ko, David N. Juurlink, Jeffrey C. Kwong.

**Project administration:** Husam Abdel-Qadir, Sarah Swayze, Michael Paterson, Dennis T. Ko, David N. Juurlink, Jeffrey C. Kwong.

**Resources:** Husam Abdel-Qadir, Sarah Swayze, Michael Paterson, Dennis T. Ko, David N. Juurlink, Jeffrey C. Kwong.

**Supervision:** Husam Abdel-Qadir.

**Visualization:** Husam Abdel-Qadir, Hardil Anup Bhatt.

**Writing – original draft:** Husam Abdel-Qadir, Hardil Anup Bhatt.

**Writing – review & editing:** Husam Abdel-Qadir, Hardil Anup Bhatt, Sarah Swayze, Michael Paterson, Dennis T. Ko, David N. Juurlink, Jeffrey C. Kwong.

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
