## [Editor Report · Decision Letter 0]

19 Mar 2025

Dear Dr Abdel-Qadir,

Thank you for submitting your manuscript entitled "The association between COVID-19 vaccination and sudden death in apparently healthy younger individuals: a population-based case-control study" for consideration by PLOS Medicine.

Your manuscript has now been evaluated by the PLOS Medicine editorial staff as well as by an academic editor with relevant expertise and I am writing to let you know that we would like to send your submission out for external peer review.

Please re-submit your manuscript within two working days, i.e. by Mar 21 2025 11:59PM.

Kind regards,

Suzanne De Bruijn, PhD

Associate Editor

PLOS Medicine

---

## [Decision Letter · Decision Letter 1]

28 May 2025

Dear Dr Abdel-Qadir,

Many thanks for submitting your manuscript "The association between COVID-19 vaccination and sudden death in apparently healthy younger individuals: a population-based case-control study" (PMEDICINE-D-25-01000R1) to PLOS Medicine. The paper has been reviewed by subject experts and a statistician; their comments are included below and can also be accessed here: [LINK]

As you will see, although the reviewers outline several concerns, they support a revision. We strongly recommend that you follow the suggestions of reviewer #2 and adjust the analyses accordingly. After discussing the paper with the editorial team and an academic editor with relevant expertise, I'm pleased to invite you to revise the paper in response to the reviewers' comments. We plan to send the revised paper to some or all of the original reviewers, and we cannot provide any guarantees at this stage regarding publication.

We ask that you submit your revision by Jun 18 2025. However, if this deadline is not feasible, please contact me by email, and we can discuss a suitable alternative.

Don't hesitate to contact me directly with any questions (sbruijn@plos.org).

Best regards,

Alexandra Tosun, PhD

atosun@plos.org

[on behalf of]

Suzanne De Bruijn, PhD

Associate Editor

PLOS Medicine

sbruijn@plos.org

Comments from the academic editor:

The Academic Editor thinks that you could state more clearly what is novel about your study. They agree with the comments made by reviewers #2 and #3 regarding residual confounding and encourage you to follow reviewer #2's advice and adjust the analyses.

Comments from the reviewers:

Reviewer #1: This population-based case-control study has large sample size, reliable data source, the selection of cases is clear and well-explained, the statistical approach is straightforward, and the results are well organised, for example, it is nice to have Table 1 where the readers can have a clear understanding of the cases' demographic information as compared with the general population.

Minor comments:

This study has a large dataset to closely match the cases by age, sex and residence region. However, the matching by age is not clear. As by Line 134, "Controls were matched to their case according to date of birth, sex, and forward sortation area", controls should have the exact age as that of the cases. However, the variable age differs between case and control groups in the tables (Table 1, 2, S3 ). Please explain. If the matching is not limited to the exact date of birth, please clarify. If a case has more than 5 potential controls, what are the principles to select 5 from them? Are there cases with less than 5 matched controls?

Table 1 contains 4963 cases while Table 2 contains 4806 cases. Can the authors explain why over 150 cases were removed?

The COVID pandemic triggered many abnormalities in public health and medical settings. It would be interesting to see the longitudinal trend of the sudden death rate (same definition of cases in this study) before and during COVID, and for during COVID, before and after Vaccine initiation. Having those figures - without any statistical inference - would be the icing on the cake.

The term "undoubtedly" in line 303 is too strong.

Reviewer #2: This paper addresses an important safety concern expressed largely in social media and other non-scientific fora that COVID-19 vaccines can cause sudden death in previously healthy young people. The authors use a matched case control design to test the hypothesis of an increased risk of death after COOVID-19 vaccine and find no evidence in support of this. I have the following comments:

Line 187 says that prior to matching the cases were older and more likely to be male but doesn't say than whom. In table 1 the comparator column is called controls but they are just the non-cases in the eligible population. I think it is confusing to label them controls. Since this is by design a matched case control study I am unclear as to the relevance of the detail shown in Table 1. If the authors think table 1 is important to show, perhaps they can discuss its relevance otherwise it would seem more appropriate to show it in Supplementary appendix for background information.

The numbers of cases in Table 2 which is with matched controls has dropped to 4896 from 4936 in Table 2. The authors should account for this difference.

Despite restricting the population under study by a number of factors that would predispose towards sudden death and matching controls on date of birth, sex and first 3 digits of the postcode, a lower proportion of cases than controls had received influenza vaccination. This would suggest residual confounding in some health care seeking or other behaviour/risk factors which would not necessarily be fully adjusted for by including receipt of influenza vaccine in the regression model. The consistency of the findings of an apparent protective effect of vaccination, with a lower OR after two than one dose is consistent with there being an unmeasured confounding variable associated with health care seeking behaviour. Given this, it would have been informative to conduct an analysis using the self-controlled cases only method to see whether the apparent protective effect was still there with a method that automatically controls for time invariant individual level confounding variables. Did the authors consider this method which is now widely used for vaccine safety studies to deal with the problem of unmeasured confounding? In any event this issue should be discussed by the authors and while they authors are correct that their study did not provide evidence of an increased risk of sudden death a small elevated risk may have been masked by the confounding.

Line 273. The authors say that "Our data align with recent publications that also report no association of COVID-19 vaccination with sudden death". They cite one study from England that used the SCCS method (ref 37) but this study found no evidence of a protective effect either. Neither did a separate study in England (Stowe J et al. Risk of cardiac arrhythmia and cardiac arrest after primary and booster COVID-19 vaccination in England: A self-controlled case series analysis. Vaccine X. 2023 Dec 1;15:100418 - not referenced by the authors). In contrast a case control study from India (ref 39) did find an apparent protective effect of COVID-19 vaccination. It would have been valuable if the authors had considered potential reasons for the apparent protective effect in the two case-control studies and why this was not observed in the two self controlled cases only studies. At the end of the discussion (line 317) the authors do seem to acknowledge the potential for confounding by saying that unvaccinated individuals may be less likely to seek health care encounters and to be more likely to have undiagnosed health conditions but don't then discuss what impact they think this this might have on their study results.

Reviewer #3: The authors present a case control study of death and COVID-19 vaccination. The case-control study design is an appropriate choice. Potential confounders have been considered, and measured confounders have been accounted for by matching or adjustment. The paper is well written.

The protective effect of COVID-19 vaccination found could suggest that there remains unmeasured confounding, e.g. as a result of differential reporting of COVID-19 infection among cases and controls, but of course it is impossible to be certain. It is noted as a limitation in the discussion that rapid antigen tests could not be captured and so unmeasured infection could not be accounted for (this would be an issue for any study design).

I agree with the overall conclusion: There is no evidence from this study that COVID-19 vaccination is associated with an increased risk of cardiac death. Given the potential for unmeasured confounding, I think that the overall conclusions are fair and are not overstated.

---

* Please upload any figures associated with your paper as individual TIF or EPS files with 300dpi resolution at resubmission; please read our figure guidelines for more information on our requirements: http://journals.plos.org/plosmedicine/s/figures. While revising your submission, please upload your figure files to the PACE digital diagnostic tool, https://pacev2.apexcovantage.com/. PACE helps ensure that figures meet PLOS requirements. To use PACE, you must first register as a user. Then, login and navigate to the UPLOAD tab, where you will find detailed instructions on how to use the tool. If you encounter any issues or have any questions when using PACE, please email us at PLOSMedicine@plos.org.

* ETHICS STATEMENT: Please include the approval number and details on consent procedures (also, if waived).

FIGURES AND TABLES

SUPPLEMENTARY MATERIAL

REFERENCES

STUDY TYPE-SPECIFIC REQUESTS

* Abstract: Please include the study design, population and setting, number of participants, years during which the study took place (enrollment and follow up), length of follow up, and main outcome measures.

* Please ensure that the study is reported according to the RECORD guideline (available from https://www.record-statement.org) and include the completed checklist as Supporting Information. If you believe that the STROBE guideline is more appropriate, please use it. Please add the following statement, or similar, to the Methods: "This study is reported as per the Reporting of Studies Conducted using Observational Routinely-Collected Data (RECORD) guideline (S1 Checklist)." When completing the checklist, please use section and paragraph numbers, rather than page numbers.

* For all observational studies, in the manuscript text, please indicate: (1) the specific hypotheses you intended to test, (2) the analytical methods by which you planned to test them, (3) the analyses you actually performed, and (4) when reported analyses differ from those that were planned, transparent explanations for differences that affect the reliability of the study's results. If a reported analysis was performed based on an interesting but unanticipated pattern in the data, please be clear that the analysis was data driven.

* Please state in the Methods section whether the study had a prospective protocol or analysis plan. If a prospective analysis plan (from your funding proposal, IRB or other ethics committee submission, study protocol, or other planning document written before analyzing the data) was used in designing the study, please include the relevant document(s) with your revised manuscript as a Supporting Information file to be published alongside your study and cite it in the Methods section. A legend for this file should be included at the end of your manuscript. If no such document exists, please make sure that the Methods section transparently describes when analyses were planned, and when/why any data-driven changes to analyses took place. Changes in the analysis, including those made in response to peer review comments, should be identified as such in the Methods section of the paper, with rationale.

---

## [Decision Letter · Decision Letter 2]

17 Dec 2025

Dear Dr. Abdel-Qadir,

Thank you very much for re-submitting your manuscript "The association between COVID-19 vaccination and sudden death in apparently healthy younger individuals: a population-based case-control study" (PMEDICINE-D-25-01000R2) for review by PLOS Medicine.

I have discussed the paper with my colleagues and the academic editor and it was also seen again by 2 reviewers. I am pleased to say that provided the remaining editorial and production issues are dealt with we are planning to accept the paper for publication in the journal.

Before we can accept your manuscript, we have several editorial requests we need you to address:

A) We would like you to address all remaining concerns from Reviewer 2, including modifying the abstract.

B) We would like you to reframe your narrative in the introduction, as well as in the rest of the manuscript, to not base your research on a social media narrative. We would suggest to just state that 'there is a fear that the vaccine leads to sudden death despite lack of evidence'. Please also remove the specific examples calling people out by name. Similarly, please rephrase the bullet point in the authors summary under 'what do these findings mean' to just state your conclusion, rather than that this 'refutes a narrative'.

[LINK]

We look forward to receiving the revised manuscript by Jan 07 2026 11:59PM.

Sincerely,

Suzanne De Bruijn, PhD

Associate Editor

PLOS Medicine

plosmedicine.org

Requests from Editors:

GENERAL

* Please confirm that your title complies with PLOS Medicine's style. Your title must be nondeclarative and not a question. It should begin with main concept if possible. "Effect of" should be used only if causality can be inferred, i.e., for an RCT. Please place the study design ("A randomized controlled trial," "A retrospective study," "A modelling study," etc.) in the subtitle (ie, after a colon).

* Please ensure that all abbreviations are defined at first use throughout the text.

* Please confirm that all numbers presented in the abstract are present and identical to numbers presented in the main manuscript text.

* Please replace "subject" with participant, patient, individual, or person.

FUNDING STATEMENT

* The funding statement should include: specific grant numbers, initials of authors who received each award, URLs to sponsors’ websites. Also, please state whether any sponsors or funders (other than the named authors) played any role in study design, data collection and analysis, the decision to publish, or preparation of the manuscript. If they had no role in the research, include this sentence: “The funders had no role in study design, data collection and analysis, decision to publish, or preparation of the manuscript.”

* Please add URLs to the funders website, as well as grant numbers where possible.

COMPETING INTERESTS STATEMENT

* All authors must declare their relevant competing interests per the PLOS policy, which can be seen here: https://journals.plos.org/plosmedicine/s/competing-interests For authors with ties to industry, please indicate whether any of the interests has a financial stake in the results of the current study.

ETHICS

• Include IRB approval number

• Was informed consent necessary, or was this waived? Please state so in the methods section. If it was necessary, please state whether informed consent was written or oral.

ABSTRACT

* Please confirm that your abstract complies with our requirements, including format (three sections: Background, Methods and Findings, and Conclusions) and providing all the information relevant to this study type https://journals.plos.org/plosmedicine/s/submission-guidelines#loc-abstract

* In the abstract, please include the important dependent variables that are adjusted for in the analyses.

AUTHOR SUMMARY

* In the author summary, in the final bullet point of 'What Do These Findings Mean?', please include the main limitations of the study in non-technical language.

TABLES

* When a p value is given, please specify the statistical test used to determine it in the legend.

* Please remove the p-values from the tables describing baseline characteristics.

DISCUSSION

* Please remove the 'conclusions' subheading from the discussion. Please also remove any other subheadings from the discussion.

OBSERVATIONAL, COHORT, CROSS-SECTIONAL, AND CASE CONTROL STUDIES

* Did your study have a prospective protocol or analysis plan? Please state this (either way) early in the Methods section.

* Please include in the methods section that the SCCS sensitivity analysis was added after reviewer requests.

* For all observational studies, in the manuscript text, please indicate: (1) the specific hypotheses you intended to test, (2) the analytical methods by which you planned to test them, (3) the analyses you actually performed, and (4) when reported analyses differ from those that were planned, transparent explanations for differences that affect the reliability of the study's results. If a reported analysis was performed based on an interesting but unanticipated pattern in the data, please be clear that the analysis was data-driven.

Comments from Reviewers:

Reviewer #1: The authors have clearly addressed my previous questions; the manuscript is well-written, and I agree with the study methodology and their conclusions.

Reviewer #2: The revisions the authors have made are comprehensive and address all my comments. The addition of the SCCS analysis is particularly informative as unlike the matched case control study it shows no evidence of a protective effect of COVID-19 vaccination which is consistent with my concerns that the protective effect in the case control analysis reflected residual confounding. This point is acknowledged by the authors on lines 322-324 in the discussion. I am therefore surprised that the authors have modified the conclusion of their abstract which now states that "COVID-19 vaccination was consistently associated with a lower risk of sudden death". What is the justification for adding "consistently" when the SCCS did not show a protective effect especially as there are now grounds for concluding that the apparent protective effect from the case control study likely reflects residual confounding? A more balanced conclusion would be to omit this sentence and just say "These data do not support the hypothesis that COVID-19 vaccines increase the risk of sudden cardiac death" which was the final sentence in the authors' original abstract. This would also bring the conclusion in the abstract in line with the Author Summary (lines 77-81).

Additional comments:

I was unable to find mention of what the acronym ICES stands for in the paper. Three different organisations came up with this acronym in Canada in my wed search. There is a link to how to access ICES data in the data sharing section of the paper but it would be helpful if the name of ICES was given in full with a generic weblink to the main ICES website in the reference list for readers to understand exactly what this organisation is and how it works.

Line 336 - is there a verb missing here eg "found"

[LINK]

---

## [Editor Report · Decision Letter 3]

13 Jan 2026

Dear Dr. Abdel-Qadir,

Thank you very much for re-submitting your manuscript "The association between COVID-19 vaccination and sudden death in apparently healthy younger individuals: a population-based case-control study" (PMEDICINE-D-25-01000R3) for review by PLOS Medicine.

We have a few remaining editorial requests:

1) Please confirm that your abstract complies with our requirements, including format (three sections: Background, Methods and Findings, and Conclusions) and providing all the information relevant to this study type https://journals.plos.org/plosmedicine/s/submission-guidelines#loc-abstract

2) Please ensure that all abbreviations are defined at first use throughout the text.

3) Please confirm that all numbers presented in the abstract are present and identical to numbers presented in the main manuscript text.

FUNDING STATEMENT

4) Please include initials of authors who received each award.

5) Please add URLs to the funders website, as well as grant numbers where possible.

ETHICS

6) Include IRB approval number

7) Was informed consent necessary, or was this waived? Please state so in the methods section. If it was necessary, please state whether informed consent was written or oral.

OTHER

8) Please mention the fact that differences in health care seeking behaviours may lead to residual confounding as an additional limitation in the abstract.

9) Line 104: include ‘previously’ to make clear these data are not presented in this manuscript.

10) Line 109: add a sentence stating the knowledge gap: e.g. ‘sudden death in apparently healthy people has not been studied yet’.

11) Line 328: remove ‘decisively’

12) We appreciate that you discuss the differences in covid vaccinations after matching in detail (line 250-256); however, please also mention the difference in covid vaccines in the sentence prior to this paragraph, to avoid any possible confusion. (line 242-244: “Most differences between cases and controls were nullified, but there remained a higher prevalence of hypertension and mood/anxiety disorders, and lower documented influenza vaccination, among cases.”)

We look forward to receiving the revised manuscript by Jan 20 2026 11:59PM.

Sincerely,

Suzanne De Bruijn, PhD

Associate Editor

PLOS Medicine

plosmedicine.org

---

## [Editor Report · Decision Letter 4]

22 Jan 2026

Dear Dr Abdel-Qadir,

On behalf of my colleagues and the Academic Editor, Rebecca Grais, I am pleased to inform you that we have agreed to publish your manuscript "The association between COVID-19 vaccination and sudden death in apparently healthy younger individuals: a population-based case-control study" (PMEDICINE-D-25-01000R4) in PLOS Medicine.

Before your manuscript can formally we accepted, we have two remaining request:

1) Please remove 'speculated' from the sentence on line 112.

2) Please consider removing 'the' from your title.

Furthermore, before your manuscript can be formally accepted you will need to complete some formatting changes, which you will receive in a follow up email. Please be aware that it may take several days for you to receive this email; during this time no action is required by you. Once you have received these formatting requests, please note that your manuscript will not be scheduled for publication until you have made the required changes.

PRESS

Sincerely,

Suzanne De Bruijn, PhD

Associate Editor

PLOS Medicine